# The Significant Role of PA28αβ in CD8^+^ T Cell-Mediated Graft Rejection Contrasts with Its Negligible Impact on the Generation of MHC-I Ligands

**DOI:** 10.3390/ijms25115649

**Published:** 2024-05-22

**Authors:** Katharina Inholz, Ulrika Bader, Sarah Mundt, Michael Basler

**Affiliations:** 1Biotechnology Institute Thurgau (BITg) at the University of Konstanz, 8280 Kreuzlingen, Switzerland; katharina.inholz@uni-konstanz.de; 2Division of Immunology, Department of Biology, University of Konstanz, 78464 Konstanz, Germany; 3Institute of Experimental Immunology, University of Zurich, 8057 Zurich, Switzerland

**Keywords:** proteasome, PA28αβ, CD8 T cell response, adoptive cell transfer, rejection, antigen processing, antigen presentation

## Abstract

The proteasome generates the majority of peptides presented on MHC class I molecules. The cleavage pattern of the proteasome has been shown to be changed via the proteasome activator (PA)28 alpha beta (PA28αβ). In particular, several immunogenic peptides have been reported to be PA28αβ-dependent. In contrast, we did not observe a major impact of PA28αβ on the generation of different major histocompatibility complex (MHC) classI ligands. PA28αβ-knockout mice infected with the *lymphocytic choriomeningitis virus* (*LCMV*) or *vaccinia* virus showed a normal cluster of differentiation (CD) 8 response and viral clearance. However, we observed that the adoptive transfer of wild-type cells into PA28αβ-knockout mice led to graft rejection, but not vice versa. Depletion experiments showed that the observed rejection was mediated by CD8^+^ cytotoxic T cells. These data indicate that PA28αβ might be involved in the development of the CD8^+^ T cell repertoire in the thymus. Taken together, our data suggest that PA28αβ is a crucial factor determining T cell selection and, therefore, impacts graft acceptance.

## 1. Introduction

The ubiquitin–proteasome system is the main producer of peptides presented on MHC class I molecules. The differently expressed catalytic active subunits in the standard proteasome (β1, β2 and β5), the immunoproteasome (β1i, β2i and β5i) and the thymoproteasome (β1i, β2i and β5t) lead to an altered MHC-I peptide repertoire in different tissues [1,2,3,4,5,6,7]. In addition, the binding of different regulatory particles to the core 20S proteasome shapes the peptide repertoire [8,9]. The most common bound activator is the 19S regulator, which drastically increases the catalytic activity of the 20S proteasome and forms the 26S proteasome by binding to the 20S core particle. Additionally, there are other activators able to bind to the 20S core particle, for example, the proteasome activator 28 alpha beta (PA28αβ). In contrast to the 19S activator, PA28αβ can shape the peptide generation of its substrates in an ATP independent process [10]. PA28αβ is composed of three alpha and four beta subunits, forming a ring-like structure [11]. It can bind at both sides of the proteasome or build a hybrid proteasome together with one 19S regulator [12,13]. This hybrid proteasome has been shown to hydrolyze the substrate into rather short and hydrophilic peptides at a faster rate than the 26S proteasome. Interestingly, although PA28αβ was discovered to strongly activate the hydrolysis of short fluorogenic peptide substrates by the 20S proteasome, the protein turnover of PA28αβ bound to the 20S proteasome alone was shown to be as slow as for the 20S proteasome [14]. Although there are several hypotheses how PA28αβ shapes and changes peptide production, the underlying mechanisms are not fully understood. PA28αβ might serve as a sieve, leading to a longer stay of peptides within the 20S proteasome and, therefore, favoring the production of short peptides [15]. Alternatively, it has been discussed that by binding of PA28αβ to the 20S proteasome, the catalytic activity of the 20S subunits might be changed [14].

Both subunits of PA28αβ are induced by the inflammatory cytokine interferon (IFN)-γ. Due to this, and the fact that it can bind to both the immunoproteasome as well as the standard proteasome with the same affinity, a role of PA28αβ in immunological processes has been proposed [15,16]. Indeed, an impact of PA28αβ on the generation of *murine cytomegalovirus* (*MCMV)*, *coxsackievirus B3* and *influenza A virus* pathogen-derived peptides has been reported [17,18,19]. However, the biological relevance in immunity seems rather low because PA28αβ-knockout cells in mice showed a mild impact of PA28αβ on the generation of immunogenic peptides [20,21].

Although the vast majority of peptides produced by PA28αβ-capped proteasomes bound to the proteasome are too small for efficient MHC class I presentation [14], a role for PA28αβ in the generation of self-peptides has been proposed [22]. The proteasome seems to play a pivotal role in the selection of cytotoxic T cells in the thymus [1,2,23,24,25,26]. The unique thymoproteasome shapes the peptide repertoire, which is presented in cortical thymic epithelial cells (cTECs) for the positive selection of CD4^+^CD8^+^ thymocytes [27,28]. Subsequent to the positive selection, the remaining T cells undergo negative selection, which is based on the presentation of peptides mostly generated by the immunoproteasome and expressed by medullary thymic epithelial cells (mTECs) and dendritic cells (DCs). PA28αβ is expressed in the thymus [26] and, therefore, has the possibility to bind to the thymoproteasome or the immunoproteasome, contributing to the peptide repertoire for T cell selection. However, if proteasome activators such as PA28αβ indeed contribute to the selection of T cells in the thymus in addition to shaping the peptide repertoire presented in the periphery remains to be investigated.

In this study, we re-investigated the contribution of PA28αβ to MHC-I antigen processing and viral infection in vivo in PA28αβ-deficient mice.

## 2. Results

### 2.1. PA28αβ Does Not Influence the Distribution of Immune-Cell Subsets or the MHC Class I Expression in Naïve Mice

To determine if PA28αβ could influence the composition of immune cells in the spleen and thymus, PA28αβ-deficient mice and C57BL/6 control mice were stained for CD4, CD8, CD3, CD19, NK1.1, CD11b and CD11c and analyzed by flow cytometry. The assessment of these immune-cell population did not reveal any impact attributable to the absence of PA28αβ (Figure 1a,b). Given the important role of the proteasome in the MHC class I presentation of immunogenic peptides, we investigated the potential impact of PA28αβ on the MHC class I surface expression. Murine embryonic fibroblasts (MEFs) derived from either wild-type mice or PA28αβ-deficient mice were treated with IFN-γ. The H-2K^b^ surface expression was analyzed by flow cytometry. IFN-γ induced an upregulation of H-2K^b^ in a time-dependent manner. However, the MHC class I surface expression remained unchanged in the absence of PA28αβ (Figure 1c). Comparative analyses of the most prevalent immune-cell subsets in the spleen between wild-type and PA28αβ-knockout mice revealed no significant differences in the MHC class I surface expression on T cells (CD3^+^), cytotoxic T cells (CD8^+^), T helper cells (CD4^+^), B cells (CD19^+^) or natural killer cells (NK1.1^+^) (Figure 1d). Our findings suggest that the potential contribution of PA28αβ to the processing of immunogenic peptides is rather negligible.

### 2.2. Presentation of Several Tested Immunogenic MHC Class I Peptides Remains Unaltered by PA28αβ Expression

It has been described that PA28αβ increases the MHC class I presentation of some immunogenic peptides [18,29,30], whereas others remain unaffected [21,31]. To address if PA28αβ affected further T cell epitopes, five different H-2D^b^/K^b^-restricted *LCMV*-derived epitopes were investigated. MEFs from either wild-type or PA28αβ-knockout mice were infected with *LCMV-WE* and the amount of peptides presented on MHC class I molecules was detected using peptide-specific CTL lines against glycoprotein (GP) or nucleoprotein (NP) NP_396–404_/Db, GP_276–286_/Db, GP_33–41_/Db, GP_118–125_/Kb and NP_205–212_/Kb. Similar to the *LCMV*-derived NP_118-126/_Ld epitope [30], PA28αβ had no significant effect on the presentation of different *LCMV*-derived peptides (Figure 2a–e). Externally, peptide-loaded MEFs were used as positive controls (Appendix A).

Next, we investigated the presentation of ovalbumin-derived SIINFEKL. rVV-Ova-infected MEFs from wild-type or PA28α^−/−^β^−/−^ mice were infected with recombinant *vaccinia* viruses expressing ovalbumin (rVV-OVA). The SIINFEKL presentation was determined using SIINFEKL-specific B3Z T cell hybridoma activation. The peptide presentation was independent of PA28αβ, confirming previous results (Figure 2f) [20]. Furthermore, the presentation of *influenza*-derived NP_147-155_/Ld was shown to depend on PA28αβ [29]. To re-investigate the presentation of this peptide, PA28αβ-overexpressing cell lines (BPαβm2 and BPαβ13) and their parental cell lines C4 and B8 were infected with a *vaccinia* virus expressing *influenza*-NP (rVV-PR8-NP). The amount of NP_147-155_ peptides presented on MHC class I molecules was detected using peptide-specific CTL lines (Figure 2g). No difference in NP_147-155_ presentation between cells overexpressing PA28αβ (BPαβ2 and BPαβ13) and cells with low PA28αβ (C4 and B8) could be detected. Taken together, our data investigating 7 different MHC-I peptides suggest a rather negligible impact of PA28αβ on the generation and presentation of immunogenic peptides.

### 2.3. Immune Response after Vaccinia or LCMV Infection Is Not Altered in PA28α^−/−^β^−/−^ Mice

The peptide presentation on MHC-I is a crucial step in the induction of a cytotoxic T cell response. Here, we used a *vaccinia* or an *LCMV-WE* infection model system to address whether PA28αβ influenced the presentation in vivo, thereby altering the induction of a cytotoxic T cell response. Wild-type or PA28αβ-deficient mice were infected with VV-WR and viral titers were determined in the ovaries on day 6 post-infection. *Vaccinia*-infected PA28αβ-knockout mice did not show altered viral titers in the ovaries 6 days after infection compared with infected wild-type mice (Figure 3a). An analysis of the cytotoxic T cell response on day 8 post-infection to four different *vaccinia*-derived peptides did not reveal any differences (Figure 3b). Next, wild-type or PA28αβ-deficient mice were infected with *LCMV-WE*. Viral titers were determined in the spleen on day 2, 4, 6 and 8 post-infection (Figure 3c). Similar viral titers were observed in wild-type and PA28αβ-deficient mice. The viral clearance in *LCMV*-infected C57BL/6 mice was strictly dependent on cytotoxic T cells [32]. Mice lacking PA28αβ were able to eliminate *LCMV* within 8 days, indicating that these mice were fully competent in mounting a cytotoxic T cell response. Indeed, the CTL response in PA28αβ-knockout mice and wild-type mice directed against GP_33–41_/Db/Kb, NP_396–404_/Db, GP_276–286_/Db, GP_92–101_/Db, GP_118–125_/Kb and NP_205–212_/Kb on day 8 post-*LCMV-WE* infection by intracellular cytokine staining (ICS) for IFN-γ showed a similar strength and hierarchy (Figure 3d). Our results, derived from two different viral infection models, were in agreement with a rather mild impact of PA28αβ on the induction of CD8^+^ T cells after *Listeria monocytogenes* infection in mice [20].

The in vivo CTL response (Figure 3) was in line with the in vitro antigen-presentation assays (Figure 2) as both datasets showed that PA28αβ had no impact on the presentation of the investigated MHC-I ligands.

### 2.4. Thymocyte Development and TCR Subsets Remain Unaltered by PA28αβ Deficiency

As PA28αβ is expressed in the thymus [26] and because the proteasome plays an important role in the selection of CD8^+^ T cells, we investigated whether PA28αβ influenced the development of T cells in the thymus. We checked the CD44^+^/CD25^+^ expression, which is used to differentiate between CD4^−^/CD8^−^ double-negative (DN) 1, DN2, DN3 and DN4 thymocyte populations. No discernible differences were observed in the DN development between wild-type and PA28αβ-knockout mice (Figure 4a). Additionally, the CD5 expression, serving as a marker for the intensity of T cell receptor (TCR) interactions with self-MHC peptides, and the CD69 expression, indicating the successful positive selection of T cells, were evaluated across CD4^+^ single positive (SP), CD8^+^ SP, and double-positive (DP) thymocytes in both wild-type and PA28αβ-deficient mice. The expression levels of CD5 and CD69 on all examined subsets did not vary, which indicated an unimpeded T cell development in PA28αβ-deficient mice (Figure 4b). Furthermore, the TCR composition was examined using a flow cytometry TCR Vβ screening panel for both CD8^+^ splenocytes and CD8^+^ thymocytes from wild-type and PA28αβ-deficient mice (Figure 4c,d and Appendix A for CD4^+^ splenocytes and CD4^+^ thymocytes). The TCR Vβ analysis revealed no differences in either the bulk T cell selection in the thymus or in the periphery in the spleen of PA28αβ-deficient mice. In addition, the H-2D^b^ and H-2K^b^ surface expression on mTECs and cTECs was analyzed in the thymus of wild-type mice and PA28αβ-deficient mice (Appendix A). Neither the total amount of mTECs or cTECs nor their MHC-I surface expression were different in PA28αβ-deficient mice.

### 2.5. PA28αβ-Deficient Mice Reject Transplanted Wild-Type Cells in a CD8^+^ T Cell-Dependent Manner

The proteasome is an attractive target and a crucial factor determining transplantation efficiency [33,34,35,36]. Hence, it seems obvious that PA28αβ, as a regulator of the proteasome, might influence transplantation efficiency. Therefore, we performed adoptive cell transfer experiments. Magnetically sorted wild-type CD3^+^ CD45.1^+^ T cells were transferred into PA28αβ-knockout mice (expressing the CD45.2 isoform) and vice versa. In weekly assessments, the survival of the transferred T cells was analyzed in the blood by flow cytometry (Figure 5a–d). We observed a rejection of wild-type cells in PA28αβ-knockout mice, while transferred wild-type cells remained stable in wild-type mice over five weeks. An analysis of the spleen after five weeks post-transfer showed that wild-type T cells were completely rejected (Figure 5b). Interestingly, wild-type mice receiving T cells from PA28αβ-knockout mice did not reject the transferred cells (Figure 5c,d).

To gain insight into the underlying mechanism of the rejection of wild-type T cells transferred to PA28αβ-knockout mice, we depleted the CD8^+^ T cells (Appendix A) in the PA28αβ-knockout host mice before and during the transfer of wild-type CD19^+^ B cells. To avoid the antibody-mediated depletion of transferred T cells in the host mice, we used magnetically sorted CD19^+^ B cells for the transfer experiments. Similar to T cells (Figure 5a,b) wild-type B cells were rejected in PA28αβ-deficient recipient mice. Interestingly, transferred wild-type B cells were not rejected in CD8 T cell-depleted host mice. These data show that the rejection of wild-type cells in PA28αβ-deficient mice is mediated by cytotoxic T cells. 

Consequently, the decreased survival of adoptively transferred wild-type cells into PA28αβ-deficient mice prompted us to investigate the influence of PA28αβ in vivo using a murine skin transplantation model.

For this purpose, C57BL/6 or PA28α^−/−^β^−/−^ tail-skin grafts were transplanted onto the back of PA28αβ-deficient mice or wild-type mice, respectively. Skin transplantation from wild-type mice to wild-type mice was used as a negative control. The graft on the recipient mice was monitored daily for signs of rejection. Both the wild-type and PA28αβ-deficient mice did not show any signs of graft rejection within 11 weeks (Appendix A). These data show that PA28αβ had no influence on skin transplantation in the used model. It has been reported that immunoproteasome-deficient mice might harbor an altered T cell repertoire [2,23,24,25,26]. Furthermore, the ubiquitin-like modifier FAT10, which targets proteins for proteasomal degradation, has been shown to modify the T cell repertoire [37]. Hence, as the proteasome regulator PA28αβ is also expressed in the thymus [26], an altered T cell repertoire in PA28αβ-deficient mice might be responsible for the rejection of transferred wild-type cells in PA28αβ-deficient mice (Figure 5). To investigate if the PA28αβ expression altered the T cell repertoire in the thymus, thereby being responsible for the rejection of transferred wild-type cells, we decided to use bone-marrow chimeras. Wild-type or PA28αβ-deficient mice were irradiated and reconstituted with either wild-type or PA28αβ-deficient bone marrow. We intended to transfer magnetically purified cells 8 weeks post-reconstitution in a similar experiment as performed in Figure 5. If wild-type cells transferred to PA28αβ-deficient mice reconstituted with wild-type bone marrow were still rejected, then PA28αβ was required in radioresistant cells—most likely, thymic epithelial cells. However, within two weeks post-radiation, most of the PA28αβ-deficient mice reconstituted with wild-type bone marrow had to be euthanized due to severe health conditions. As the PA28αβ mice reconstituted with PA28αβ bone marrow were healthy, the PA28αβ-deficient mice were not susceptible to radiation per se. Although these data further strengthen the conclusion that the PA28αβ expression is an important parameter in transplantation, additional experiments need to be performed to elucidate the underlying mechanisms of the rejection of wild-type bone marrow in PA28αβ-deficient mice.

## 3. Discussion

In the present study, we re-investigated the role of PA28αβ in the generation of MHC class I peptides and furthermore investigated its role in viral infections and its impact on tissue grafting.

The fact that PA28αβ is IFN-y-inducible and binds to the proteasome led to the hypothesis that PA28αβ was involved in MHC class I antigen processing. Peptide generation studies have proposed that the binding of PA28αβ to the proteasome leads to the release of smaller and more hydrophilic peptides [14]. Different in vitro and in vivo studies have shown an impact of PA28αβ on the generation of some MHC-I ligands [31,38]. Furthermore, it has been shown that in PA28αβ-deficient cells, the IFN-γ-inducible MHC class I expression was reduced, which indicates an impact of PA28αβ on the presentation of ligands upon IFN-γ stimulation [39]. Yet, the biological relevance remains unclear. In contrast to these studies, we investigated several MHC-I ligands using in vitro antigen-presentation assays and could not confirm a major impact of PA28αβ on the generation of immunogenic MHC-I peptides (Figure 2). Additionally, we did not observe a difference in MHC-I upregulation in IFN-γ-stimulated wild-type and PA28αβ-deficient MEFs (Figure 1c). In agreement, we did not observe an impact of PA28αβ on the induction of a cytotoxic T cell response in two different viral models in vivo. This was in line with initial findings that PA28αβ-knockout mice did not show a difference in body-weight loss and mortality after *influenza* infection [31]. Furthermore, Sijts et al. showed a negligible role of PA28αβ in the *Listeria monocytogenes*-specific CD8 and CD4 response eight days after infection [20]. In line with these results, we showed that PA28αβ has no crucial role in the antiviral response after *LCMV* or *vaccinia* virus infections. No difference in the cytotoxic T cell response and viral titers was observed (Figure 3). Interestingly, there are indications that in COVID-19 patients, SARS-CoV-2 infection can trigger PA28αβ upregulation [40]. Whether this upregulation is just a result of increased IFN-γ or whether PA28αβ has additional beneficial effects on the infection outcome itself remain to be investigated.

Our results from adoptive cell transfer experiments suggest a role of PA28αβ in the generation of self-peptides. PA28αβ expression in the thymus indicates that by binding to the thymoproteasome in cTECs, it can shape the generation of peptides responsible for the positive selection of CD8^+^ cells [26]. Additionally, the DCs, mTECs and stromal cells responsible for negative selection in the thymus express immunoproteasomes, which are capable of binding to PA28αβ. This indicates that PA28αβ might be involved in the generation of the peptides responsible for negative selection in the thymus [3,26]. We could not detect any differences between the abundance of mTECs and cTECs or any differences in the surface expression of MHC class I peptides on mTECs and cTECs when comparing wild-type and PA28αβ-knockout mice (Appendix A). This strengthens the hypothesis that the observed rejection of wild-type cells in knockout mice relies on the role of PA28αβ in shaping the peptide repertoire present in the TECs rather than a bulk MHC-I expression.

Thus, the absence of PA28αβ in the thymus may lead to an altered MHC-I peptidome, positively and negatively selecting cells in the thymus compared with wild-type cells. The selection processes in the presence of new or missing self-peptides in PA28αβ-deficient mice can lead to an altered T cell repertoire. This might result in the observed depletion of transferred wild-type cells in knockout mice as some peptides presented on wild-type MHC class I are recognized as foreign peptides. This notion is strongly supported by the fact that cytotoxic T cells were responsible for the rejection of transferred cells in our model (Figure 5c). The production of low-affinity peptides for MHC class I presentation is crucial for positive selection by cTECs. The ability of PA28αβ to shape a peptide repertoire that is too short for an efficient MHC presentation might have an impact on the positive selection of T cells by increasing the pool of low-affinity self-peptides [14]. Thus, a lack of PA28αβ could result in an altered TCR diversity. Furthermore, a PA28αβ deficiency in mTECs and DCs in the thymus might lead to missing self-peptides, which are required for the negative selection of cytotoxic T cells. This is important in the context of transplantation as T cells leave the thymus able to trigger an allo-response after transplantation when organs or cells from a healthy donor are transplanted. A potential change in T cell clonality, which might trigger rejection after the transfer of cells, was not observed in our flow cytometry studies on thymocyte development, TCR usage and TCR subsets in PA28αβ-knockout mice. This implies that PA28αβ does not alter the bulk MHC-I presentation in the thymus, but that differences are probably related to a few peptides. To detect these subtle differences, an analysis of the TCR subsets with more sensitive methods such as TCR sequencing might reveal PA28αβ-induced alterations in the T cell repertoire. The clonal expansion of a few T cells after transplantation is sufficient for a graft rejection. These clones display only a small proportion compared with the bulk T cells and might easily remain undetected by flow cytometry Vβ screening (Figure 4) [41].

Investigating the tissue-specific peptidome in allo-recognition has been shown to be highly important to identify self-peptides impacting graft survival [42]. To further elaborate the involvement of PA28αβ in graft rejection, we performed tail-skin transplantation experiments. We observed no rejection of skin transferred from wild-type mice onto PA28αβ-knockout mice (Appendix A). Interestingly, skin transferred from wild-type mice onto LMP7^−/−^ mice was rejected within 2 to 6 weeks in earlier studies [33], indicating that the immunoproteasome, compared with PA28αβ in the thymus, has a broader effect on MHC-I ligand processing and, thus, the T cell repertoire. The 26S and the PA28αβ hybrid proteasomes interact with different proteins [9,43]; therefore, some MHC-I ligands can be over-represented, leading to recognition by and rejection of specific cytotoxic T cells. Such peptides might be differently expressed on transferred immune cells compared with transplanted skin. Furthermore, the PA28αβ expression in transferred immune cells and skin might be different. As the transplanted skin needs to undergo vascularization, which takes several days to weeks, an acute rejection is less common in this model and might be responsible for the difference in the rejection of skin and transferred cells [44]. The possibility that peptides originating from PA28αβ itself were present in the cells of wild-type mice cannot be excluded. Nevertheless, as PA28αβ is expressed in the skin, we would have expected a rejection in the skin transplantation model, which we did not observe. The rejection of transferred cells but not skin is, therefore, based on the impact of PA28αβ on peptide generation rather than being present itself on MHC.

Our attempt at using bone-marrow chimeras to corroborate that selection processes in the thymus were the underlying mechanism for the rejection of transferred immune cells failed due to the immediate rejection of the wild-type bone marrow in the PA28αβ-knockout mice. Despite not being able to investigate the impact of PA28αβ-dependent thymic selection processes in bone-marrow chimeras, this outcome further highlighted the importance of PA28αβ in graft acceptance. As proteasomal activators have a major impact on the catalytic activity of the proteasome, they must be taken into consideration before and after transplantations [45].

We concluded that the role of PA28αβ in the generation of immunogenic MHC-I ligands has been overestimated. Putative alterations in MHC-I processing had no biological relevance in our two viral infection models. PA28αβ was not required to conduct a potent immune response after viral infections and eliminate the viruses. Strikingly, PA28αβ was found to be an essential factor in graft acceptance.

## 4. Materials and Methods

### 4.1. Cell Lines

Mouse embryonal fibroblasts derived from either C57BL/6 or PA28α^−/−^β^−/−^ were produced as described previously and maintained in a DMEM medium (Gibco, Fisher Scientific, Schwerte, Germany) supplemented with 10% FCS (Gibco, Fisher Scientific, Schwerte, Germany) and 100 U/mL penicillin/100 µg/mL streptomycin (Gibco, Fisher Scientific, Schwerte, Germany) [46]. The cell lines C4 [30], B8 [47], BPαβ13 and BPαβ2 [30] and the hybridoma cell line B3Z [48] were maintained in IMDM (Gibco, Fisher Scientific, Schwerte, Germany) supplemented with 10% FCS and 100 U/mL penicillin/100 µg/mL streptomycin. BSC-40 and MC57 cell lines were cultured in MEM (Gibco, Fisher Scientific, Schwerte, Germany) supplemented with 5% FCS and 100 U/mL penicillin/100 µg/mL streptomycin. Adherent cells were detached by incubation with 0.05% trypsin-EDTA (Gibco, Fisher Scientific, Schwerte, Germany) for 5 min at 37 °C. All cell lines were maintained at 37 °C with 5% CO_2_ in a humidified atmosphere.

### 4.2. Mice and Ethical Statement

C57BL/6 (H-2b) and B6.SJL-PtprcaPep3b/BoyJ (Ly5.1 congenic mice) mice were originally purchased from Charles River, Sulzfeld, Germany. PA28α^−/−^β^−/−^ mice (B6.Cg-Psme1/Psme2tm1Tchi) were kindly provided by T. Chiba (Department of Molecular Oncology, Tokyo Metropolitan Institute of Medical Science, Tokyo, Japan) [31]. All animals were bred in air-conditioned rooms with a controlled temperature of 21 °C, 55% relative humidity and constant ventilation (17 air changes/h) in the animal facility of the University of Konstanz under pathogen-free conditions with a 12 h light/dark cycle. Animals were provided ad libitum access to a standard animal diet and water. For the infection models, the mice were infected either with 200 pfu *LCMV-WE* i.v. or with 2 × 10^6^ pfu VV-WR i.p. for the indicated time points. The animal experiments were approved by the Review Board of Governmental Presidium Freiburg (G-20/071, G-23/023, G-23/027, I-18/03 and I-22/001).

### 4.3. Antigen-Specific CTL Lines

CTL lines were generated as previously described [2]. In brief, C57BL/6 mice and PA28α^−/−^β^−/−^ mice were either infected with 200 pfu *LCMV-WE* i.v. or with 2 × 10^6^ pfu recombinant *vaccinia* i.p. (rVV-PR8-NP) viruses. Four weeks after infection, single-cell suspensions of splenocytes were incubated with 40 U/mL IL-2 and incubated with 10^−5^ M of the corresponding peptide for 8 days. IL-2 was freshly added every second day. Dead cells were removed after 8 days via Ficoll-Paque^TM^ (Merck, Darmstadt, Germany) gradient centrifugation. Viable cells were incubated with the corresponding peptide and IL-2 for another 7–10 days. A second Ficoll-Paque^TM^ gradient was performed one day prior to the experiment.

### 4.4. Antigen-Presentation Assay and Hybridoma Assay

For the *LCMV* antigen-presentation assays, C57BL/6 and PA28α^−/−^β^−/−^ MEFs were stimulated with 100 U/mL IFN-γ for 48 h. Afterwards, cells were infected with *LCMV-WE* (MOI 0.5) for 24 h at 37 °C. Viable peptide-specific CTLs were incubated with MEFs for 5 h with the addition of 10 μg/mL Brefeldin A (Thermo Fisher Scientific, Darmstadt, Germany). Antigen presentation was analyzed via intracellular staining for the IFN-γ^+^ of CD8^+^ CTLs. For the SIINFEKL presentation, C57BL/6 and PA28α^−/−^β^−/−^ MEFs were stimulated with 100 U/mL IFN-γ for 48 h followed by three hours of infection with rVV-Ova (MOI 10). Next, 5 × 10^5^ MEFs were co-cultured with 10^6^ B3Z hybridomas for 24 h at 37 °C. The activation of hybridomas was analyzed via an IL-2 ELISA according to the manufacturer’s protocol (mouse IL-2 ELISA Ready-SET-Go!^®^, eBiosciences,, Frankfurt a. Main, Germany). For influenza A peptide generation (NP_147-155_), C4, B8, BPαβ13 and BPαβm2 were infected with rVV-PR8-NP_147-155_ (MOI 10) for 4.5 h. Subsequently, the infected cells were co-cultured with peptide-specific CTLs for three hours. Antigen presentation was analyzed via the intracellular staining of IFN-γ^+^ CD8^+^ CTLs as previously described [25].

### 4.5. mTEC and cTEC Preparation

Thymi of three to four week old mice were cut into small pieces and digested with 0.25 mg/mL collagenase D (Roche, Mannheim, Germany), 1 U/mL dispase I (Gibco, Fisher Scientific, Schwerte, Germany) and 25 µg/mL DNase I (Roche, Mannheim, Germany) in RPMI supplemented with 2% FCS and 25 mM HEPES at 37 °C using a gentleMACS (C Tubes, Miltenyi, Miltenyi Biotec GmbH, Bergisch Gladbach, Germany, Program 37_ABDK_1). Single-cell suspensions were stained for the flow cytometry analysis.

### 4.6. Flow Cytometry

Single-cell suspensions from the spleen and thymus or MEFs were labeled with fluorochrome-conjugated antibodies in a FACS buffer (2% FCS, 2 mM EDTA and 2 mM NaN_3_ in 1x PBS) for 20 min at 4 °C. For intracellular staining, cells were fixed with 4% paraformaldehyde and permeabilized using a FACS buffer containing 0.1% saponin (Quillaja sp., Sigma-Aldrich, Darmstadt, Germany). Flow cytometry was performed using a BD FACSLyric™ instrument (BD Biosciences, Heidelberg, Germany). The following antibodies were purchased from the indicated companies and used in the indicated concentrations: CD3ε-APC (145-2C11, BioLegend, Koblenz, Germany; 1:250), CD4-APC (GK1.5, eBioscience™, Frankfurt a. Main, Germany; 1:250), CD4-PE (GK1.5, BioLegend, Koblenz, Germany; 1:250), CD4-BV510 (GK1.5, BioLegend, Koblenz, Germany; 1:250), CD5-PE (53-7.3, BD Pharmingen™, Heidelberg, Germany; 1:250), CD8a-APC (53-6.7, BioLegend, Koblenz, Germany; 1:250), CD8a-PE (53-6.7, BioLegend; Koblenz, Germany; 1:250), CD11b-FITC (M1/70, eBioscience™, Frankfurt a. Main, Germany; 1:250), CD11b-PE (M1/70, eBioscience™, Frankfurt a. Main, Germany; 1:250), CD19-FITC (eBio1D3, eBioscience™, Frankfurt a. Main, Germany; 1:250), CD25-PECy7 (3C7, BioLegend, Koblenz, Germany; 1:250), CD44-APC Fire Cy750 (IM7, BioLegend; 1:250), CD45.1-APC (A20, Miltenyi Biotec GmbH, Bergisch Gladbach, Germany; 1:40), CD45-BV421 (30-F11, BioLegend, Koblenz, Germany; 1:250), CD45.2-PE (104-2, Miltenyi Biotec GmbH, Bergisch Gladbach, Germany; 1:40), CD69-BV421 (H1.2F3, BioLegend, Koblenz, Germany; 1:250), B220-PerCP (RA3-6B2, BD Pharmingen™, Heidelberg, Germany; 1:250), IFN-γ-FITC (XMG1.2, BioLegend, Koblenz, Germany; 1:150), NK1.1-PE (PK136, BD Pharmingen™, Heidelberg, Germany; 1:250), EpCAM-APC (G8.8, eBioscience™, Frankfurt a. Main, Germany; 1:250), Ly-51-FITC (6C3, BD Pharmingen™, Heidelberg, Germany; 1:250) and TCR Vβ Screening Panel (BD Pharmingen™, Heidelberg, Germany; concentration as indicated in the protocol).

### 4.7. Adoptive Cell Transfer

CD3^+^ cells were purified using a CD3ε MicroBead Kit (Miltenyi Biotec GmbH, Bergisch Gladbach, Germany) and CD19^+^ using a Pan B Cell Isolation Kit II (Miltenyi Biotec GmbH, Bergisch Gladbach, Germany) according to the manufacturer’s protocols. Next, 1 × 10^7^ cells in 100 µL PBS were injected i.v. Blood was taken once a week for five weeks and analyzed for CD3^+^ or CD19^+^ cells, respectively. CD8 depletion was performed by an i.p. injection of a CD8α-depleting antibody (250 µg, purified from clone YTS 169.4.2) once a week, starting one day prior to the adoptive cell transfer. The depletion efficiency was checked via flow cytometry. The cell transfer of cells from male mice into female mice was used as a control for rejection and is depicted in Appendix A.

### 4.8. Skin Transplantation

Skin transplantation experiments were performed as previously described [49]. Briefly, the skin of the tails of sacrificed donor mice was transplanted onto the back of recipient mice. After bandage removal, graft survival was monitored over five weeks.

### 4.9. LCMV and Vaccinia Titer Determination

The *LCMV* titer was titrated on MC57 cells as previously described [50]. The *vaccinia* titer was titrated on BSC-40 cells. Next, 1.5 × 10^5^ BSC-40 cells were seeded the day before by adding the organ lysates into a 24-well plate. The organ samples were prepared as a ten-fold serial dilution and 200 μL of the lysate or the dilutions were added to the BSC-40 cells. One hour after the addition of the lysates and dilutions, 1 mL MEM was added and the cells were incubated for 24 h. Afterwards, the medium was aspirated and the plaques were stained with 0.5% (*w*/*v*) crystal violet for 20 min. The crystal violet was washed away and the plaques were counted and calculated according to the dilution.

### 4.10. Statistics

The statistical analysis was performed using a two-tailed Student’s *t*-test, a one-way or a two-way ANOVA followed by a Bonferroni post hoc or Tukey’s test for a comparison of multiple groups. Unless noted differently in the figure legends, data from at least three experiments or *n* ≥ 3 were presented as the mean ± SEM. The statistical analyses were performed using GraphPad Prism software (version 9.5.1, GraphPad Software, Inc., La Jolla, CA, USA).

## Figures and Tables

**Figure 1 ijms-25-05649-f001:**
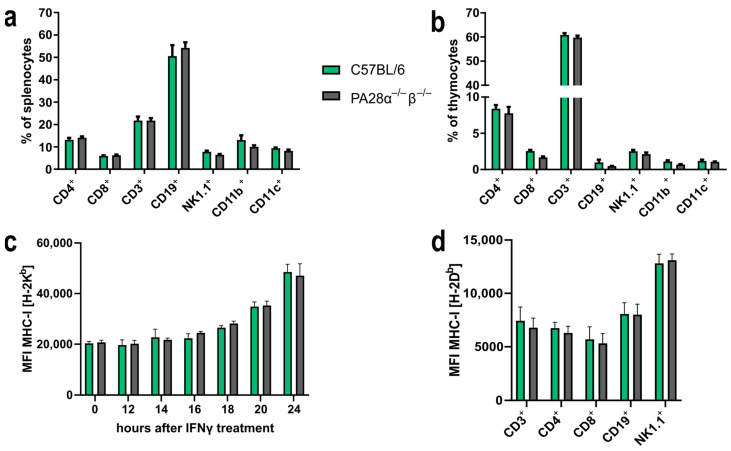
PA28αβ expression alters neither the distribution of immune-cell subsets in the spleen and thymus nor the MHC class I expression of surface on MEFs or splenocytes. (**a**,**b**) Percentage of indicated immune-cell subsets in the spleen (**a**) or thymus (**b**) was measured by flow cytometry from naïve wild-type or PA28αβ-knockout mice (*n* = 5 mice per group, measured in duplicate). (**c**) Median fluorescence intensity (MFI) of surface MHC class I H-2K^b^ expression of MEF populations from either PA28αβ-knockout or wild-type mice measured by flow cytometry after IFN-γ treatment for the indicated time points. Data show two independent MEF populations per wild-type and knockout measured as duplicates in two independent experiments. (**d**) MFI of surface expression of MHC class I H-2D^b^ measured by flow cytometry on indicated immune-cell populations in the spleen of wild-type or PA28αβ-knockout mice (*n* = 5 mice per group, measured in duplicate). (**a**–**d**) Wild-type mice are represented by green bars and PA28αβ-knockout mice are represented by grey bars. Data are shown as mean ± SEM, statistically analyzed using the Student’s *t*-test. Unless otherwise specified, the analyses revealed no statistically significant differences.

**Figure 2 ijms-25-05649-f002:**
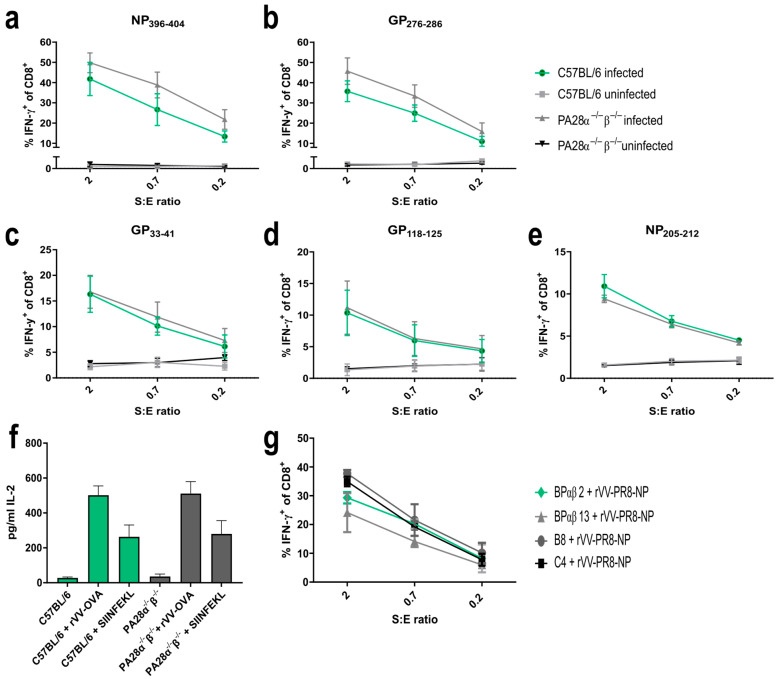
Expression of PA28αβ does not change the presentation of several *LCMV*-derived MHC-I peptides, SIINFEKL or Influenza-NP_147-155_. (**a**–**e**) IFN-γ-stimulated MEFs (48 h) from three individual preparations of C57BL/6 or two individual preparations of PA28α^−/−^β^−/−^ mice were *LCMV-WE*-infected for 24 h and afterwards co-cultured with indicated peptide-specific CTL lines at different S:E ratios (stimulators: MEFs; effector: CTL). Activation of CTL lines was analyzed by staining for CD8 and intracellular IFN-γ after five hours of co-culture. Shown are the percentages of IFN-γ-positive cells of CD8+ cells as determined by flow cytometry for 3–5 independent experiments per peptide. (**f**) MEFs from C57BL/6 or PA28α^−/−^β^−/−^ mice were infected with rVV-OVA for 3 h and SIINFEKL presentation was analyzed using B3Z T cell hybridoma. IL-2 secretion was measured by ELISA. Externally, SIINFEKL peptide-loaded MEFs (indicated as SIINFEKL) were used as positive controls (*n* = three independent experiments). Wild-type mice are represented by green bars and PA28αβ-knockout mice are represented by grey bars. (**g**) The parental cell lines C4 and B8 (low PA28αβ expression) and the cell lines BPαβ13 and BPαβ2 (high PA28αβ expression) were infected with rVV-PR8 for 3 h and co-cultured with a NP_147-155_-specific CTL line. After 4.5 h of co-culture, activation of CTL lines was analyzed by staining for CD8 and intracellular IFN-γ. Shown are the percentages of IFN-γ-positive cells of CD8^+^ cells as determined by flow cytometry. (**a**–**g**) Data are shown as mean ± SEM, statistically analyzed using the Student’s *t*-test or by a two-way ANOVA using Tukey’s multiple comparisons test. Unless otherwise specified, the analyses revealed no statistically significant differences.

**Figure 3 ijms-25-05649-f003:**
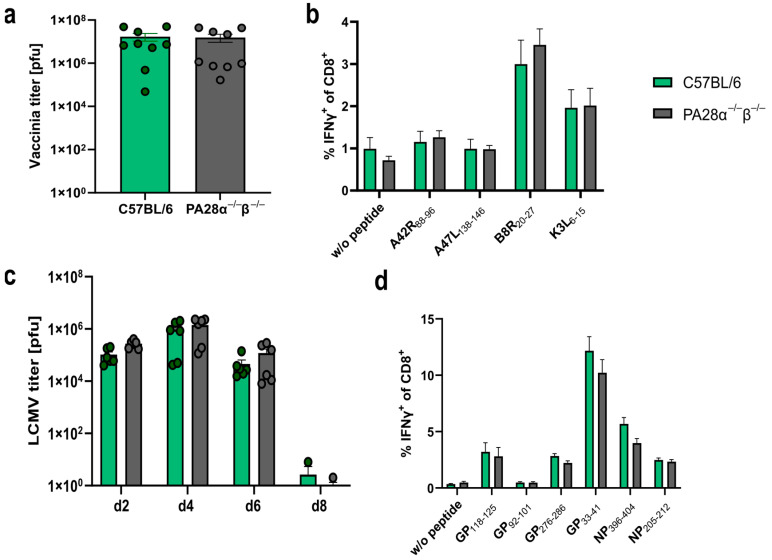
*Vaccinia* and *LCMV* titers as well as the cytotoxic T lymphocyte (CTL) response are not altered in PA28α^−/−^β^−/−^ mice. (**a–d**) Wild-type and PA28αβ-knockout mice were infected with VV-WR (**a**,**b**) or *LCMV-WE* (**c**,**d**). (**a**) Viral titers analyzed in ovaries on day 6 post VV-WR infection (*n* = 9 mice per group, represented by green dots for wild-type mice and grey dots for PA28αβ-knockout mice). (**b**,**d**) Eight days post-infection, spleen cells were harvested and stimulated in vitro with the indicated *vaccinia* virus- (**b**) or *LCMV*-derived (**d**) peptides for 5 h, and analyzed by flow cytometry after staining for CD8 and intracellular IFN-γ. Unstimulated cells (indicated by w/o peptide) were used as a negative control (*n* = 9 mice per group). (**c**) Viral titers analyzed in the spleen on indicated days post-*LCMV-WE* infection (*n* = 3–6 mice per group, represented by green dots for wild-type mice and grey dots for PA28αβ-knockout mice). (**a–d**) Wild-type mice are represented by green bars and PA28αβ-knockout mice by grey bars. Data are shown as mean ± SEM, statistically analyzed using the Student’s *t*-test. Unless otherwise specified, the analyses revealed no statistically significant differences.

**Figure 4 ijms-25-05649-f004:**
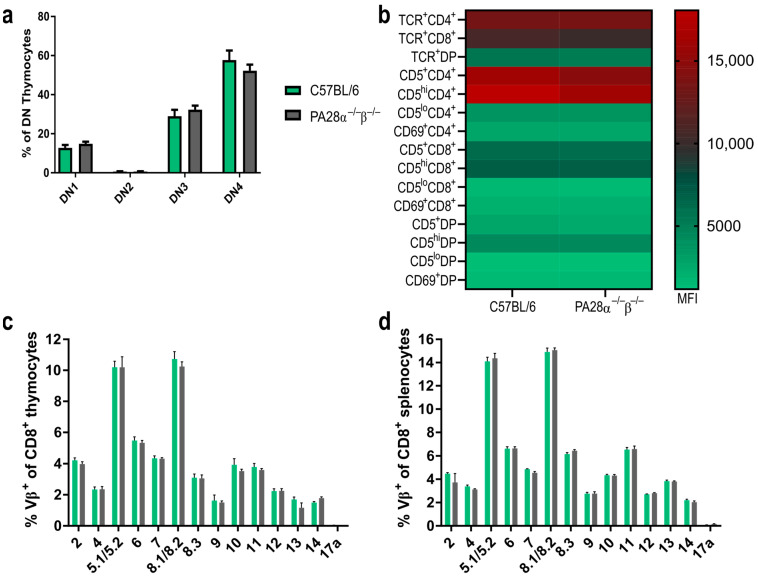
Thymocyte development and TCR distribution is not altered in the absence of PA28αβ. (**a**) DN (CD4^−^/CD8^−^) populations in the thymus of wild-type mice or PA28αβ-deficient mice were investigated according to their CD44 and CD25 expression. DN1: CD44^+^CD25^−^; DN2: CD44^+^CD25^+^; DN3: CD44^−/lo^CD25^+^; DN4: CD44^−/lo^CD25^−^ (*n* = 10 mice per group). (**b**) Median fluorescence intensity (MFI) of TCR, CD5 or CD69 on CD4 SP (CD4^+^/CD8^−^) (indicated as CD4^+^), CD8 SP (CD4^−^/CD8^+^) (indicated as CD8^+^) and DP (CD4^+^/CD8^+^) (indicated as DP) thymocytes was measured for *n* = 10 mice per group of wild-type or PA28αβ-deficient mice. (**c**,**d**) Flow cytometric analysis of indicated Vβ variable segments of TCRs from CD8 SP thymocytes (**c**) or splenocytes. Wild-type mice are represented by green bars and PA28αβ-knockout mice are represented by grey bars. (**d**) Derived from C57BL/6 mice or PA28αβ-deficient mice (*n* = 5 mice per group). Vβ17a was not expressed in the C57BL/6 background and was used as a negative control. (**a**–**d**) Data are shown as mean ± SEM, statistically analyzed using the Student’s *t*-test. Unless otherwise specified, the analyses revealed no statistically significant differences.

**Figure 5 ijms-25-05649-f005:**
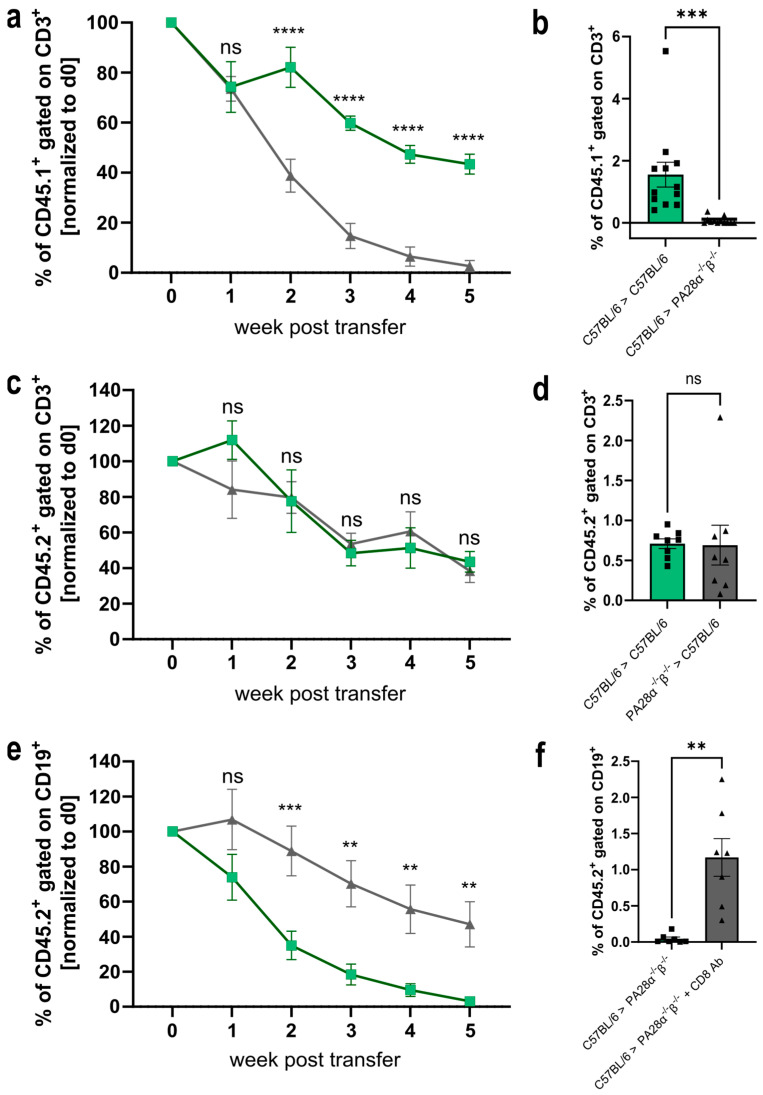
Wild-type cells are rejected in PA28αβ-knockout mice in a CD8^+^ T cell-dependent manner. (**a**,**c**) Magnetically sorted CD3^+^ T cells from wild-type (CD45.1^+^) or PA28αβ-knockout mice (CD45.2^+^) were transferred into PA28αβ-knockout or wild-type mice. The rejection of the transferred cells was monitored in the blood weekly. Green lines indicate the transfer of wild-type cells into wild-type mice, while grey lines show the transfer of wild-type cells into PA28αβ deficient mice (a) or PA28αβ deficient cells into wild-type mice (c). (**b**,**d**) Five weeks after the transfer, the percentage of CD45.1^+^ or CD45.2^+^, respectively, was measured in the spleen. (**e**) MACS-sorted CD19^+^ B cells were transferred into PA28αβ-knockout mice, which were either treated weekly with an anti-CD8^+^-depleting antibody (indicated as CD8 Ab) or left untreated. Rejection of the transferred cells was monitored in the blood weekly. Green lines indicate the transfer of wild-type cells into PA28αβ deficient mice, while grey line indicate the transfer of PA28αβ deficient cells into wild-type mice treated with an CD8 Ab. (**f**) Five weeks after the transfer, the percentage of CD45.2^+^ was measured in the spleen (*n* = 7–12 mice per group). (**a–f**) Cell transfers between mice of the same gender were performed. The transfer of cells between donors and recipients of different genders is shown in Appendix A. (**b**,**d**,**f**) Recipient wild-type mice are represented by green bars and recipient PA28αβ-knockout mice are represented by grey bars. Data statistically analyzed using a two-way ANOVA with Tukey’s multiple comparisons test. ** *p* < 0.01; *** *p* < 0.001; **** *p* < 0.0001. ns: Not significant.

## Data Availability

The data that support the findings of this study are available from the corresponding author upon reasonable request.

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
