# Peer review of "The Significant Role of PA28αβ in CD8^+^ T Cell-Mediated Graft Rejection Contrasts with Its Negligible Impact on the Generation of MHC-I Ligands"

_ijms, 2024, doi:10.3390/ijms25115649_

Round 1

Reviewer 1 Report

Comments and Suggestions for Authors

Proteolytic activity of proteasomes is fine-tuned by diverse regulatory complexes that associate with the 20S proteasome. While proteasomes play an essential role in MHC class I antigen processing, it is unclear to which extent the regulatory complexes contribute to antigen presentation and thereby to pathogen-specific immunity. In this manuscript, Inholz et al. revisit the potential contribution of the proteasome regulatory complex PA28αβ to MHC class I antigen presentation and immune responses, using double gene-deficient mice that lack PA28α and PA28β expression. In line with previous observations, presented data fail to show any differences in immune cell distribution in spleen and thymus, MHC class I expression levels, antiviral immune responses, and thymic T cell development in PA28αβ-deficient compared to wt mice. However, it appears that adoptively transferred wt T and B cells are rejected by PA28αβ-deficient mice, which is mediated by CD8 T cells. In addition, a larger percentage of PA28αβ-deficient mice cannot be reconstituted with wt bone marrow transplants, while wt mice accept PA28αβ-deficient bone marrow and accept transferred  PA28αβ-deficient cells. Based on their data and a paper by the group of Niedermann (Eur. J. Immunol. 2004) showing PA28αβ expression in both cTECs and mTECs, the authors suggest that subtle differences in MHC class I presentation of self-peptides in the PA28αβ-deficient mouse’ thymus may affect CD8 T cell selection and thereby impact graft acceptance.

In general, although especially the argumentation in the discussion (line 321-325) on the potential impact of PA28αβ on cTEC-mediated positive selection is highly interesting, the suggested role of PA28αβ in T cell selection is insufficiently demonstrated in this manuscript.

In the first place, the authors should consider the possibility that PA28 may act as an antigen on transferred wt cells.

In addition, the materials and methods lack any details on potential sex differences between donor and acceptor mice (although Fig. S2a however suggests that expression of the male H-Y antigen was considered in the experimental set up).

The bone marrow chimera experiments are more puzzling than in favor of the manuscript’s conclusion. In general, BM transplants with altered MHC class I peptide presentation, for example bm1 BM in B6 mice and vice versa, are tolerated. In addition, as the accepting mice must have been irradiated, it is questionable whether host CD8 T cell responses are possible within the first weeks after transplantation.

To support their conclusions, the author should provide more direct evidence for the proposed role of PA28αβ in T cell selection. For instance, the authors may examine the expression levels of MHC class I on the cell subsets responsible for T cell selection, such as mTECs, cTECs and dendritic cells. Of note, Yamano et al. (J. Exp Med 2002) already showed that IFN-γ–induced up-regulation of MHC class I is defective in PA28α/β-deficient macrophages and dendritic cells. Also the specificity of wt-specific PA28α/β-deficient CD8 T cells may be further explored and TCR usage of the TCR subsets may be determined by TCR sequencing, as suggested in line 336.

Comments on the Quality of English Language

Reviewer 2 Report

Comments and Suggestions for Authors

Inholz and colleagues provide a fairly exhaustive exploration of the impact of PA28alpha-beta on peptide presentation by MHC-I molecules. The realist are partly confirmatory but have the merit of studying multiple epitope-specific responses in vitro and in vivo as well as thyme T cell development. While most findings are negative, the authors demonstrate a strong and unidirectional CD8+ T cell-dependent rejection of wld type lymphocytes in PA28 ko mice, in addition to unidirectional fast rejection of wild type bone marrow in irradiated mice. Regarding the latter experiments, the authors should add a description of the methodology. Moreover, the mechanism of bone marrow rejection remains unclear - did the authors monitor the effect of irradiation on chimerism, e.g. using congenic mice? The authors speculate that lymphocyte and marrow rejection are related to PA38 effects in thymic selection; they might extend the discussion considering potential effects of PA28 on ligands presented by class Ib molecules (also recognized by CD8 T cells), and/or on innate immune cells and mechanisms which might be implicated in rejection of bone marrow grafts.

Reviewer 3 Report

Comments and Suggestions for Authors

In this manuscript, the authors showed that PA28ab doesn't involve in antigen presentation during viral infection, and proposed that PA28ab plays a role in self-peptide generation. Knocking out PA28ab can alter T cell repertoire and lead to depletion of adoptively transferred cells. I have a few comments: 

1. The authors claimed that PA28ab had no significant effect on the presentation of LCMV derived peptides. However, according to Figure 2a-b, presentation levels of NP364-404 and GP276-286 peptides were higher in PA28ab knock out mice. 

2. The authors only tested LCMV and ovalbumin antigens and then concluded that PA28ab was not involved in antigen presentation. However, PA28ab response may change depending on the epitope types. 

Round 2

Reviewer 1 Report

Comments and Suggestions for Authors

All comments have been adequately addressed except one, the possible rejection of wt cells in PA28 KO mice due to CD8 T cell recognition of a PA28-derived epitope. Absence of rejection in the skin transplantation model is a valid argument. Nevertheless, the authors may consider to run the PA28 sequences through an epitope prediction tool and test potential binders for recognition by spleen cells of transplanted KO mice.
